# Using the R Package Spatstat to Assess Inhibitory Effects of Microregional Hypoxia on the Infiltration of Cancers of the Head and Neck Region by Cytotoxic T Lymphocytes

**DOI:** 10.3390/cancers13081924

**Published:** 2021-04-16

**Authors:** Justus Kaufmann, Christophe A. N. Biscio, Peter Bankhead, Stefanie Zimmer, Heinz Schmidberger, Ege Rubak, Arnulf Mayer

**Affiliations:** 1Department of Radiation Oncology, University Medical Center of the Johannes Gutenberg University, 55131 Mainz, Germany; justus_kaufmann@online.de (J.K.); heinz.schmidberger@unimedizin-mainz.de (H.S.); 2Department of Mathematical Sciences, Aalborg University, Skjernvej 4A, 9220 Aalborg East, Denmark; christophe@math.aau.dk (C.A.N.B.); rubak@math.aau.dk (E.R.); 3Edinburgh Pathology, Institute of Genetics and Molecular Medicine, University of Edinburgh, Edinburgh EH4 2XR, UK; p.bankhead@ed.ac.uk; 4Institute of Pathology and Tissue Biobank, University Medical Center of the Johannes Gutenberg University, 55131 Mainz, Germany; Stefanie.Zimmer2@unimedizin-mainz.de

**Keywords:** spatial analysis, open-source, R, hypoxia, cancer immune evasion, multichannel immunofluorescence, head and neck cancer

## Abstract

**Simple Summary:**

Progress in the field of in situ proteomics allows for the simultaneous detection of multiple biomarkers within one cancer tissue specimen. As a result, biological hypotheses previously only assessable ex vivo can now be studied in human cancer tissue. However, methods for objective analysis have so far been lacking behind. In this study, we established a free, objective, and entirely open-source-based method for the analysis of multiplexed immunofluorescence specimens. This will gain further importance with the availability of more advanced multiplexing methods in the future.

**Abstract:**

(1) Background: The immune system has physiological antitumor activity, which is partially mediated by cytotoxic T lymphocytes (CTL). Tumor hypoxia, which is highly prevalent in cancers of the head and neck region, has been hypothesized to inhibit the infiltration of tumors by CTL. In situ data validating this concept have so far been based solely upon the visual assessment of the distribution of CTL. Here, we have established a set of spatial statistical tools to address this problem mathematically and tested their performance. (2) Patients and Methods: We have analyzed regions of interest (ROI) of 22 specimens of cancers of the head and neck region after 4-plex immunofluorescence staining and whole-slide scanning. Single cell-based segmentation was carried out in QuPath. Specimens were analyzed with the endpoints clustering and interactions between CTL, normoxic, and hypoxic tumor areas, both visually and using spatial statistical tools implemented in the R package Spatstat. (3) Results: Visual assessment suggested clustering of CTL in all instances. The visual analysis also suggested an inhibitory effect between hypoxic tumor areas and CTL in a minority of the whole-slide scans (9 of 22, 41%). Conversely, the objective mathematical analysis in Spatstat demonstrated statistically significant inhibitory interactions between hypoxia and CTL accumulation in a substantially higher number of specimens (16 of 22, 73%). It showed a similar trend in all but one of the remaining samples. (4) Conclusion: Our findings provide non-obvious but statistically rigorous evidence of inhibition of CTL infiltration into hypoxic tumor subregions of cancers of the head and neck. Importantly, these shielded sites may be the origin of tumor recurrences. We provide the methodology for the transfer of our statistical approach to similar questions. We discuss why versions of the Kcross and pcf.cross functions may be the methods of choice among the repertoire of statistical tests in Spatstat for this type of analysis.

## 1. Introduction

Squamous cell carcinomas of the head and neck often contain extensive tissue areas with substantially reduced oxygen partial pressures compared to the corresponding normal tissues [1]. Severe tumor hypoxia has long been recognized as a predictive factor for an inferior response to primary radiotherapy in this tumor entity [2]. Not only is the effect of radiotherapy under fully oxygenated conditions amplified by up to a factor of three (oxygen enhancement effect) compared to hypoxic/anoxic conditions, there are numerous other implications relevant to therapeutic resistance [3,4].

In recent years, the success of immune-checkpoint inhibitors has brought the immune system’s role in tumor therapy into the focus of attention. E.g., it has been shown that the immune system also partially mediates the effects of radiotherapy of cancers of the head and neck [5]. Various studies in different tumor entities have demonstrated that the density of the infiltration by CD8-positive cytotoxic T lymphocytes (CTL), and especially the pattern of this infiltration, represent vital prognostic factors [6,7].

Hypoxic tissue areas in experimental tumors have been found to contain low CTL, suggesting that hypoxic tissue regions might be partially protected from the immune system’s access [8]. Conversely, hyperoxia reactivated the immune system in the tumor microenvironment and led to the regression of metastases in experimental animals [9]. However, one difficulty in testing such a hypothesis is that the assessment of infiltration patterns is highly subjective and, therefore, associated with questionable reproducibility. Objective classifications and quantification of the tumor immune phenotype would be desirable, especially if future therapy decisions are based on such assessments.

The spatial statistics methods include tools already established in other areas of science (e.g., ecology, geosciences). They seem promising candidates to enable an objective examination of cell distribution and interaction patterns in human tissue as well. This has become possible only recently due to the availability of fluorescence slide scanners and analysis software, allowing single-cell segmentation of cells and subsequent classification into different tumor compartments (e.g., tumor cells, stromal cells) based on machine learning algorithms. In the present work, we have applied spatial statistical methods implemented in the R package Spatstat (E.R. & C.A.N.B., http://spatstat.org/) (accessed on 13 April 2021) [10] to squamous cell carcinomas of the head and neck region and compared the results with a traditional visual analysis of CTL infiltration patterns. Data acquisition in 22 specimens was performed in the open-source software QuPath (P.B., https://qupath.github.io/) (accessed on 13 April 2021) [11].

## 2. Materials and Methods

### 2.1. Selection of Specimens for the Study

A total of 103 whole-slide head and neck tumors (oral cavity, nasopharynx, oropharynx, hypopharynx, larynx) from the Biobank collection of the University Medical Center Mainz and one microarray with a total of 70 spots (US Biomax, Derwood, MD, USA, Cat.-No. HN-801a) were available for the study. Regarding the latter, some spots showed an insufficient degree of morphological preservation and could not be evaluated. Finally, we included a total of 22 specimens in the analysis, which contained both extensive hypoxic micro-regions and sufficient infiltration by CTL.

### 2.2. Immunofluorescence Staining

Slides were stained for the antigens carbonic anhydrase (CA) IX, CD8, and CD73. Only the former two antigens are relevant for this study. Results from the analysis of CD73 will be communicated soon (manuscript in preparation). The 4-plex staining protocol has been described in detail before [12]. In brief, we heated the specimens in 10/1 mm Tris/EDTA pH 9.0 buffer in a microwave oven (Bosch HMT75M451, Robert Bosch GmbH, Munich, Germany) for 15 min to retrieve antigenic binding sites. Next, we incubated tissue specimens with the primary antibodies listed in Appendix A. We used the biotin-free, micro polymer-based SuperPicture™ reagents (Thermo Fisher Scientific, Waltham, MA, USA) directed against the species of origin of the primary antibodies to label the latter with horseradish peroxidase. We achieved the visualization of antigens by incubating specimens with appropriate fluorochrome-tyramide conjugates in appropriate buffers after their activation with H_2_O_2_ (see Appendix A). We repeated microwave treatment between successive rounds of fluorescent staining to quench peroxidase activity from preceding antigen-labeling steps. In a final step, we incubated the slides with 4′,6-diamidino-2-phenylindole (DAPI) at a concentration of 300 nM. Finally, slides were covered with a coverslip using DAKO Fluorescence mounting medium (Dako Deutschland GmbH, Hamburg, Germany) and dried overnight at 4 °C.

### 2.3. Acquisition of Digital Whole-Slide Images

Slides were digitized as czi files using the Zeiss AxioScan Z.1 and appropriate single-band filter sets for the fluorescence channels, DAPI, fluorescein isothiocyanate (FITC), Cy3, and Cy5 (Morphisto GmbH, Frankfurt, Germany). For visualization and single cell-based segmentation of the digital tissue specimens, we used the open-source software QuPath (https://qupath.github.io/) (accessed on 13 April 2021) in conjunction with a built-in Bioformats extension to decode the *.czi file format.

### 2.4. Visual Assessment

Visual assessment of the specimens and the determination of the threshold levels for the hypoxia marker (i.e., hypoxic vs. non-hypoxic cells) occurred after adjustment of the gray levels to optimize visibility of individual color channels. This introduces a minor subjective component that cannot be eliminated on this technological platform. An example of a flat image without adjustments and the same image after adjustments to improve visibility is given in Appendix A. Specimens were assessed visually for clustering of CTL and subsequently categorized based upon our (J.K., A.M.) subjective estimate of the nature of the interaction between CTL and hypoxic tumor tissue into negative interaction (inhibition), no interaction, and positive interaction (clustering).

### 2.5. Data Preparation for Spatial Statistics in Spatstat

As a first step, we chose representative tissue regions of interest (ROIs) containing approximately 20,000 cells. For the whole-slide specimens, a rectangular region with a prominent expression of CA IX and CD8 was chosen. The environment of all of the ROIs is shown in Appendix A. With regard to the tissue microarray spots (n = 6), the analysis comprised the entire spot. Within these ROIs, every cell was segmented using a watershed-based algorithm implemented in QuPath. Subsequently, we programmed QuPath to classify cells into tumor and stroma using QuPath’s machine learning tools and a random-forest method. After that, we set thresholds for the antigen intensities interactively. This was necessary to account for inter-slide and inter-patient variability in fluorescence intensity resulting from differences in tissue handling (e.g., differences in fixation time). In total, we generated four classes of events corresponding to the following subpopulations of cells: 1 = CTL, 2 = normoxic tumor, 3 = hypoxic tumor, 4 = stroma. Finally, we exported the single-cell data from QuPath into a *.txt file, imported this file in R (using read.delim), and analyzed data using the R package Spatstat (http://spatstat.org/) (accessed on 13 April 2021). In Spatstat, every cell of the original image was represented as a point event in a planar (i.e., two-dimensional) point pattern belonging to an ROI derived from the original cell X-Y coordinates in the QuPath dataset. Subsequently, every event was marked using the class of the cell it represents. Figure 1 gives an overview of the entire procedure. To achieve better visibility for Figure 1, we chose a slightly smaller ROI with a lower number of cells (n = 12,635). Point patterns, which are subdivided by a categorical class variable (a factor in Spatstat terms), are referred to as multitype point patterns. Spatstat provides several tools for the analysis of the interaction between populations in multitype point patterns. Different statistical functions can be used for interaction analyses in multitype point patterns of this kind. The significance level was set to alpha = 5% for all computations. Our analysis showed that versions of the K and pcf functions might be the most suitable to the pathophysiological question of our study (see Results).

## 3. Results

Experimental studies have shown that CTL, in principle, can freely infiltrate tumor tissue, i.e., tumor and stroma areas, including the interstitial space between cells in tumor cell aggregations [13]. Figure 1A depicts an example of CTL distribution in a specimen from this study in which CTL are both observed within the tumor stroma and inside the tumor cell islets. Hence, we first asked whether CTL distribution shows systematic deviation from complete spatial randomness (CSR) in the form of clustering. Visual inspection strongly suggested that this was indeed the case in all specimens, and CTL seemed to cluster in the stroma (Figure 2).

A number of functions in Spatstat can be employed to assess the question of deviation from CSR. First, the quadrat test subdivides the ROI into a set of rectangles and tests for deviations from a homogenous intensity between them [14]. As expected, the quadrat test, in our example, indicated inhomogeneity (chi-square test, *p* < 0.0001, Figure 2A). This is important when choosing the right functions in the subsequent parts of the analysis. To adjust for the inhomogeneity shown by the quadrat test, inhomogeneous versions of the functions have to be used. These try to adjust for the inherent inhomogeneity in the pattern, which might suggest a false clustering effect otherwise. Next, we used the inhomogeneous G-function (Ginhom), which calculates the nearest neighbor probabilities, i.e., the chances of a random point of a point pattern to find its nearest neighbor within a given distance [15]. With an increasing radius around a given point in the pattern, the G-function value can only ever increase and has a theoretical maximum of 1 (i.e., 100% likelihood). When compared with the G-function of a random distribution (i.e., a Poisson point process), the G-function showed the tendency of CTL to have a higher than random chance to encounter the next CTL within a distance of 8–23 µm (Figure 2B). These analyses were repeated for the remaining 21 specimens in the study. Hence, the clustering of CTL was unanimously supported by the inhomogeneous G-function.

More advanced statistical tests enable the assessment of a deviation from spatial randomness on a continuum of radii in the two-dimensional space. The functions in Spatstat that we have employed to achieve this are Ripley’s K-function (Kest) [16] and the pair correlation function (pcf) [17,18]. Since the population of CTL was found to be distributed inhomogeneously throughout the tumors (see above), we used the inhomogeneity-corrected versions of the K-function and the pcf (Kinhom, pcfinhom) to adjust for spatial inhomogeneity [19]. As expected, we could confirm the clustering of immune cells using these functions in the previous example (Figure 2C,D). There are several reasons for this, which will be discussed (See Discussion).

To demonstrate the difference between various distribution patterns and how they might influence the K-function or the pcf, C.A.N.B. and J.K. have started to work on two Shiny-web applications that are still under further development. One showcases different patterns and their corresponding spatial statistical functions (Figure 3A–C). When comparing these different patterns to the distribution pattern of CTL, the clustering nature of CTL seemed obvious. The second app was created to provide a low-effort introduction toward experimenting with spatial statistics. Researchers can upload their own single-cell datasets and generate results. The beta versions of the apps can be accessed here (http://apps.math.aau.dk/spatstat/) (accessed on 13 April 2021).

We subsequently applied all of the previously mentioned statistical tests to the entire dataset of the study and found evidence for clustering of CTL in all cases using the inhomogeneous versions of the G-function (data not shown) and the pcf (Appendix A). To test for significance, we used the lohboot function, which calculates confidence intervals for the functions using a bootstrap method. The inhomogeneous K-function only implied clustering in 18 out of 22 cases (82%) in the study (Appendix A). This finding is most likely caused by the cumulative nature of the K-function, resulting in a tendency to underestimate the significance of a deviation from spatial randomness (see Discussion section). When all functions resulting from a single ROI are considered before making a statement about the distribution of a point pattern, there is a high implication of clustering in all specimens.

We next turned to the second and more challenging question of possible systematic avoidance of hypoxic areas by CTL. Visual inspection suggested this to be the case in only 9 of 22 patients. To assess this question using Spatstat tools, we first considered the multitype equivalent of the G-function, the Gcross function. However, we refrained from using the Gcross function for the question at hand since it is not suitable for this task due to its lack of standardization (see Discussion section). Hence, we next employed the multitype equivalents of Kest and pcf, the Kcross and pcf.cross functions using two different pairs of cell types, i.e., CTL versus normoxic (1 vs. 2), and CTL versus hypoxic tumor cells (1 vs. 3) [20,21]. Due to the inhomogeneous nature of the CTL distribution and hypoxia in the tissue, we once more employed the inhomogeneity-corrected variations. In addition, we again used the lohboot function to test the significance of possible deviations of the interaction between these pairs of cell types from complete spatial randomness. For 16 out of 22 cases, both tests showed a significant divergence of the tests’ confidence intervals for the two populations, directed in a way that implied an inhibitory effect on CTL by hypoxia. Although not statistically significant, we found trends toward a repellent effect of hypoxia on CTL in all but one of the remaining specimens (examples are shown in Figure 4).

## 4. Discussion

In the present study, we have demonstrated that CTL in squamous cell carcinomas of the head and neck region exhibit clustering and avoid hypoxic areas upon infiltration into the interstitial space between cells inside tumor cell aggregates. CA IX was chosen as an endogenous marker of hypoxic tissue areas because it has been shown to co-localize with the gold standard exogenous marker pimonidazole [22] and because the antigen has a prolonged half-life after hypoxia exposure, thereby integrating biologically relevant episodes of hypoxia over longer periods of time [23]. Since this finding has potentially broad pathophysiological significance, e.g., concerning the interaction between radiotherapy and immunotherapy, a factual methodological basis for such a statement is essential. Although the phenomenon of cluster formation seems easy to judge visually, there are borderline cases in which doubts about such behavior can arise. The expenditure for statistical test procedures in Spatstat is, as shown here, low and provides certainty in less obvious cases.

The assessment of the distribution of CTL within the tumor cell aggregates is inherently more subjective. In most specimens, it is not obvious that CTL may avoid hypoxic areas. In fact, in more than half of the cases, our visual analysis suggested that CTL did not show such behavior. It is well documented that the visual processing of patterns in the human brain is distorted by the natural attempt to generate meaning [24,25]. It may, therefore, be flawed. Specifically, there is an incentive to detect patterns consistent with a scientific study’s underlying hypothesis, even though they may not exist biologically. This is related to the phenomenon of cognitive dissonance, which describes the tendency to filter information [26]. In this setting, objective statistical analyses demonstrate their strengths and offer reproducible, investigator-independent results.

At this point, it may be essential to consider the differences between inhomogeneity and clustering. While both result in similar patterns, the reason for these patterns is vastly different. While inhomogeneity results from external factors, e.g., more favorable conditions, clustering results from internal factors or actual interaction between individual entities such as cell-to-cell-signaling. It is important to note that there is no need for direct contact between two cells to interact, as this interaction can occur via soluble factors such as chemokines. Furthermore, the scale of interaction is essential as well, e.g., when viewed at the scale of our galaxy, humans seem to show strong clustering. However, when the scale is changed to a city, then the subjective evaluation of the spatial distribution of humans is reversed. They are now more likely to be perceived as repelling each other.

In short, clustering on a large scale is more synonymous with inhomogeneity, while clustering on a small scale can be interpreted as actual clustering. Although challenging, defining the correct scale for interaction is, therefore, of utmost importance. For this study, we arbitrarily limited the size of the scale to 100 µm.

While the functions used may seem straightforward, there are a few things to be considered during the interpretation of results. The quadrat test might be a quick and easy tool to obtain a first impression of whether a point pattern is distributed randomly or not. Still, it can only detect deviations from the null hypothesis of a Poisson distribution. The cause and direction of the deviation, however, cannot be determined. Tools for additional analysis of the quadrat test may be available in the future. When analyzing the distribution of a single population, the G-function is a useful tool to detect deviations from CSR.

In the second part of our analysis, however, we compare the effects of two different populations (normoxic and hypoxic tumor cells) on a third population (CTL). To do so, we compare the results of the Cross-functions between each of the first two populations and CTL. However, the usage of the Gcross function for this question is not straightforward. It has a few pitfalls, the most important one being that it is not standardized, and therefore the results are dependent on the intensity of a point pattern. Graphs of two Gcross-functions resulting from different combinations of populations should, therefore, not be compared. Although the K-function and the pcf also have limitations, the ability to be used for inter-group comparisons is a strength of both; since they can be adjusted for the intensity of a point pattern, it is possible to compare the degree of deviation from spatial independence of the two cell populations investigated. Problems of the K-function lie within its cumulative nature. While this makes the K-function more robust for statistical inference, such as the lohboot function, it also causes a slower reaction to actual deviations from CSR present in the specimen. This can suggest that effects take place over a more extended scale of distances than is biologically the case and make them seem not as significant as they might be, resulting in a loss of statistical power on the local level. On the other hand, this cumulative nature and the more global approach make the K-function easier to estimate and, as a result, more robust for statistical inference, such as the lohboot function on a global scale.

The pcf, conversely, is not cumulative and has more local information. Compared to the K-function, it is superior when analyzing the deviations from CSR and on what scale they do take place. However, it is more difficult to estimate and may suffer from high variance, resulting in reduced statistical power globally. This can be seen in our study as well.

When evaluating the statistical results, we noticed that the inhomogeneous K-function did not significantly deviate from CSR in some of the specimens. The reasons for this are manifold. On the one hand, in specimen with a sparse immune cell infiltrate, the K-function show significantly wider confidence intervals due to smaller case numbers. Therefore, the deviation from CSR must be substantially larger to be considered significant. On the other hand, one must consider that the K-function indicates a significant departure from CSR over an entire distance.

In contrast, the pcf-function does so for a specific distance. Hence, deviations from CSR, which are recognized as significant by the pcf, do not necessarily lead to a deviation of the K-function due to an averaging effect of the entire area in which the cells are counted. Conversely, a weak effect over a longer distance may not cause the pcf to indicate a significant deviation from CSR, while the K-function may do this. Additionally, due to the cumulative nature, effects within a smaller distance may cancel out effects taking place on a bigger scale and result in the K-function not deviating from CSR.

Furthermore, attempts are made to reduce the number of false-positive results by correcting the K-function for, e.g., boundary effects and inhomogeneity. A lack of significance of the K-function does not necessarily contradict the statement that clustering occurs within a population. Looking at the curves of the corresponding pcf, however, it appears that there is only comparatively mild clustering in these particular specimens. The reasons for this could be an unfavorable environment for the CTL or other factors not tested here. Future comparisons of data from these statistical tests with survival data may identify biological differences between different types of clustering behavior of CTL [27].

While overcoming most of the subjective parts of this kind of analysis, the statistical methods presented here do not yet enable a fully automated, unbiased evaluation. Such a methodology would certainly be very interesting as it would open up the possibility of cell distribution assessment in large numbers of specimens without user interaction.

Interestingly, despite the K-function’s lack of sensitivity described above, results obtained with the cross variant of the function were surprisingly unanimous in the second part of our analysis. According to the hypothesis of hypoxia-induced immunosuppression, we expected the Kcross and pcf.cross functions to show more substantial deviations from CSR for the comparison between hypoxic tumor cells vs. CTL with normoxic tumor cells vs. CTL. Indeed, this pattern could be demonstrated for all specimens in the study (see Figure 4 and Appendix A).

Fluorescence-based multichannel staining of formalin-fixed and paraffin-embedded human tumor tissue in conjunction with whole-slide scanning, digitization, and single cell-based segmentation now offers the opportunity for mathematical analyses of human cancer tissue specimens. With the emergence of even more advanced biomarker analysis methods with the possibility of higher multiplexing, the necessity of using such methods will become increasingly apparent. Simultaneously, the data’s validity and reproducibility will improve, and the opportunity will arise to ask more complex questions than is possible with the limited number of antigens today. Four advanced systems are nanostring digital spatial profiling (nanostring DSP, [28]), imaging mass cytometry (IMC, Fluidigm [29]), multiplexed ion beam imaging (MIBI, IonPath [30]), and Cell DIVE (Cytiva, formerly General Electric [31,32]). The methodological approaches differ considerably: while Cytiva’s system uses conventional fluorescence, the two mass spectrometry techniques (IMC, MIBI) work with lanthanide-coupled antibodies, while nanostring technology uses short nucleic acid sequences that can be coupled to antibodies or RNA molecules like a barcode. Simultaneous detection of more than 30 proteins has been published for all methods, but the theoretical maximum numbers are higher. So far, the highest published number of antigens detected simultaneously was achieved on the Cell DIVE System in a study that investigated 61 antigens [32]. With the nanostring platform, however, 82 RNA targets could be detected in addition to 32 proteins.

## 5. Conclusions

In this study, we could demonstrate the value of implementing spatial statistics in the analysis of immunofluorescence-stained cancer specimens. This approach enables an objective investigator-independent in situ analysis. We could show that CTL objectively and systemically avoid hypoxic regions within cancers of the head and neck.

It is our vision that the semantics of spatial analysis in human tumor tissue, which is beginning to be established, will serve as building blocks for the spatial interpretation in the setting of far more complex investigations with high multiplexing in the future. To ease implementation of spatial statistical tools for other researchers C.A.N.B. and J.K. developed the Shiny apps mentioned above to upload single-cell data in *.csv-format for exemplary analysis (Figure 3D). The apps can be accessed here (http://apps.math.aau.dk/spatstat/) (accessed on 13 April 2021). We hope that these introductory apps may motivate researchers to use spatial statistical tools to underline statements made from in vivo observations. Such an approach may significantly deepen the pathophysiological understanding of the tumor microenvironment and serve as an in vivo benchmark for the pathophysiological concepts behind novel approaches to cancer treatment.

## Figures and Tables

**Figure 1 cancers-13-01924-f001:**
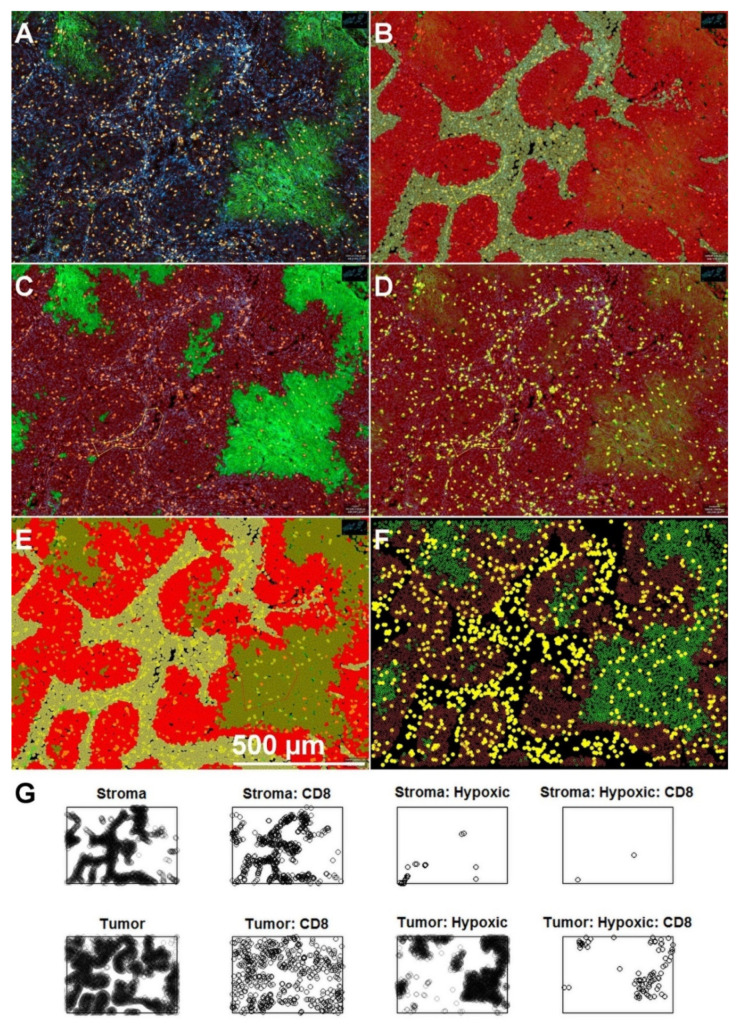
A representative area from a whole-slide specimen is shown to demonstrate the procedure of data acquisition in QuPath, cell classification, and transfer to R. (**A**) Tumor section after single-cell detection in QuPath. Approximated cell borders, extrapolated from the positions of cell nuclei, which had been segmented based on the DAPI channel (blue), are depicted by red lines. CD8-positive cytotoxic T lymphocytes (CTL) are shown in yellow. Hypoxic, CA IX-positive tumor cells are shown in green. (**B**) Result of the tumor-stroma classification using QuPath’s machine learning features. A set of example areas for tumor and stroma were delineated by the user. QuPath was then able to discriminate between both compartments for the rest of the cells in the specimen. (**C**,**D**) Construction of additional single-threshold classifiers for hypoxic cells (**C**) and CD8-positive CTL (**D**). (**E**) A compound classifier comprising tumor, stroma, and subclassifications according to the single-measurement classifiers shown in panels (**C**,**D**). Scale bar in (**E**) applies to panels **A**–**E**. (**F**) A color-coded plot of cell positions generated in R after transfer of the data and conversion to a multitype planar point pattern (ppp) in Spatstat. For the sake of clarity, subclassifications have been reorganized to CD8-positive CTL (yellow), normoxic tumor cells (brown), and hypoxic tumor cells (green). Original subclassifications are shown in panel (**G**).

**Figure 2 cancers-13-01924-f002:**
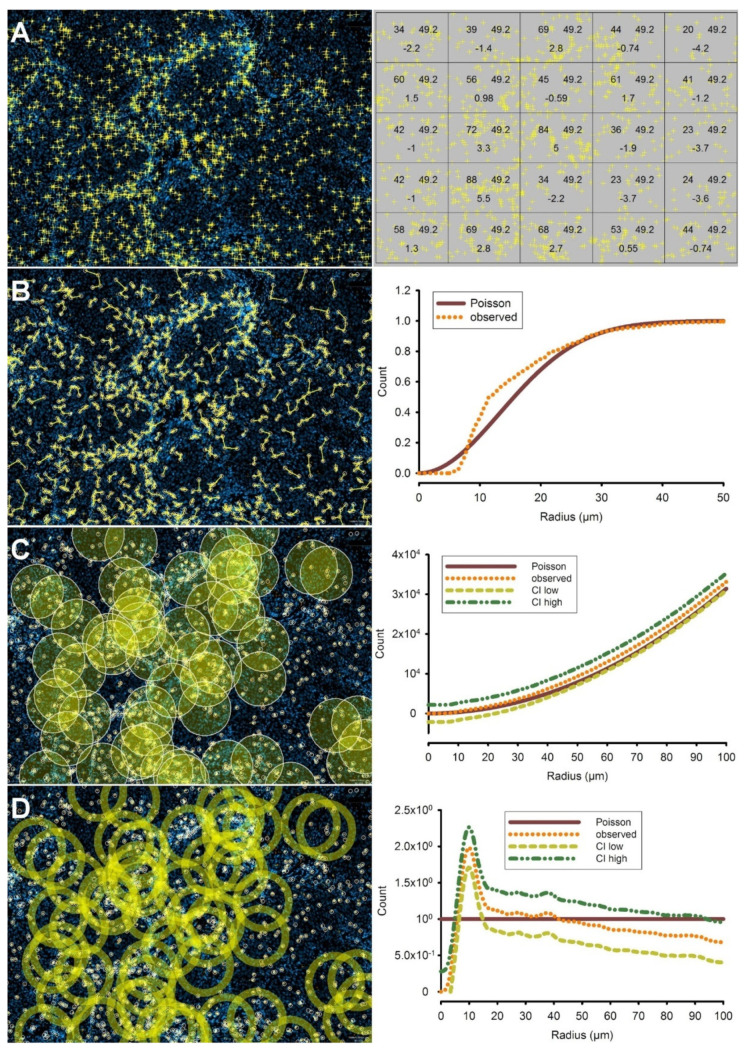
(**A**) The most simple method to assess clustering of cells is the **quadrat test**. Left panel: distribution of CTL (yellow) in the same region of a squamous cell carcinoma of the head and neck depicted in Figure 1. Clustering is less evident visually when CTL are viewed separately, outside of the context of tumor and stroma areas. First, the quadrat test (right panel) shows statistically significant inhomogeneity (*p* < 2.2 × 10^−16^). The numbers within the quadrats (from left to right) represent the number of events in each quadrat, the average number in all quadrats, and the residuals. (**B**) The **G-function** is another method to assess the clustering of CTL. **Left panel**: Graph of nearest neighbor distances of CTL (yellow lines), depicted as an overlay on the same tumor area. **Right panel**: a plot of the G-function (for any given CTL) shows a higher probability of the presence of the nearest neighboring CTL within a distance of approximately 8–23 µm than would be expected in a random distribution. The low values in the distances leading up to 8 µm are a result of the technical shrinkage of cells with a certain diameter to points for the sake of the calculation. (**C**) For the **K-function**, the average number of neighboring cell events within a circle of a given radius r is calculated for each cell event in the population. **Left panel**: representative circles of r = 50 µm are shown. This process is repeated for every value of r up to 100 µm in the region of interest (ROI). K-functions are standardized, and corrections for edge effects are performed, the details of which have been explained before. The heterogeneity of cell event numbers within these circles is evident. **Right panel**: a plot of the K-function (orange), corrected for inhomogeneity with its corresponding 95%-confidence band (green and yellow lines). Although an upward deviation from the Poisson distribution (complete spatial randomness (CSR), brown line) is evident, this did not leave the 95% confidence bands. Hence, the deviation from CSR in this example using the K-function is not statistically significant (**D**). The **pair correlation** function (pcf) of the same dataset shows pronounced clustering at low distances. **Left panel**: contrary to the K-function, the pcf analyzes the number of cells on expanding rings, thereby scanning through layers around each individual CTL. **Right panel**: Contrary to the K-function, the pcf shows significant clustering in a band between 7 and 15 µm around the cell, whereas no deviation from CSR is evident at distances beyond.

**Figure 3 cancers-13-01924-f003:**
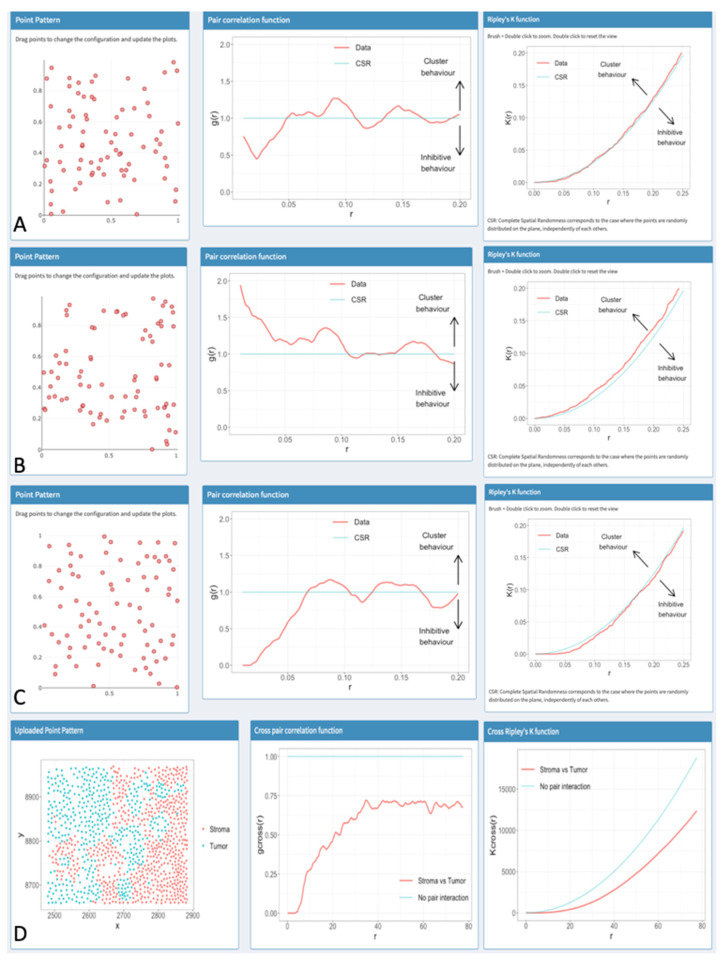
Exemplary pattern distributions and the corresponding K-function and pcf. All images in this figure were taken from the Shiny apps by C.A.N.B. and J.K. Lines corresponding to the computed functions are shown in red, while the lines corresponding to a Poisson distribution are shown in blue. (**A**) A randomly distributed pattern (Poisson distribution). While the pcf in the second column varies around the line of a theoretical Poisson distribution, the K-function of the pattern in the third column is very close to the line of a theoretical Poisson distribution, implying that the pattern might be close to complete spatial randomness (CSR). (**B**) A pattern with a clustering behavior. The corresponding pcf in the second column stays mostly above the pcf of a theoretical Poisson distribution, as does the K-function in the third column. This implies a clustering behavior. (**C**) A pattern with inhibitive behavior. The corresponding pcf in the second column stays mostly below the pcf of CSR. The K-function in the third column is below the line of CSR as well, which implies an inhibitive or regular behavior. (**D**) An exemplary dataset of the second app where tumor (blue points) and stroma (red points) cells were differentiated. Instead of the K-function and pcf used before, their Cross-functions are being used. Here one can clearly see an inhibitive or avoiding pattern where cells of one type are mostly close to each other while avoiding close contact with cells of the other type. This effect is enhanced by the inhomogeneity in the general distribution, with tumor cells being mainly on the left side of the ROI and stromal cells being almost exclusively on the right half.

**Figure 4 cancers-13-01924-f004:**
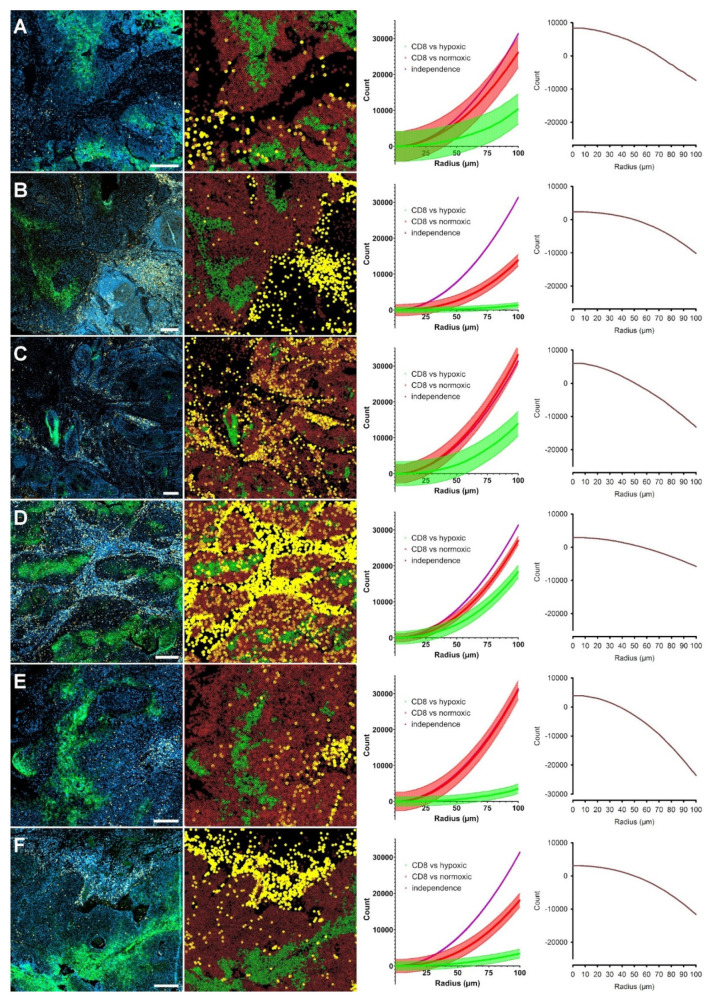
Demonstration of the repellent effect of hypoxic tissue areas on CTL infiltration. Six examples (**A**–**F**) showing significantly stronger inhibitory effects of hypoxic tumor cells compared to normoxic tumor cells concerning the distribution of CTL. Each row corresponds to one tumor. **Left panels**: original immunofluorescence images (DAPI, blue; hypoxic cells, green; CTL, yellow), all scale bars: 200 µm. **Left middle panels**, corresponding Spatstat point pattern representation of the different groups of cells identified in the tumor sections (hypoxic tumor cells, green; normoxic tumor cells, dark red; CTL, yellow). **Right middle panels**, inhomogeneity-corrected Kcross functions testing the distribution of CTL vs. hypoxic (green) and normoxic (red) tumor cells. The areas shown represent the confidence bands generated by Loh’s bootstrap method. **Right panels**, plot illustrating the delta between the lowest value of the red curve vs. the highest values of the green curves in the corresponding right middle panels. Values below 0 define a significant deviation. This is the region of statistically significant divergence between the distribution of both tumor cell populations relative to CTL. It is evident that hypoxic tumor cells have a stronger repellent or anti-clustering effect on CTL than normoxic tumor cells.

## Data Availability

Most datasets can be viewed in the app of C.A.N.B. and J.K. A total of 6 datasets from TMA-spots were excluded from the app for technical reasons (round vs. rectangular observation windows). Excluded datasets can be requested from the corresponding author. All calculated function graphs are depicted in the Appendix A.

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
