# Peer review of "Using the R Package Spatstat to Assess Inhibitory Effects of Microregional Hypoxia on the Infiltration of Cancers of the Head and Neck Region by Cytotoxic T Lymphocytes"

_cancers, 2021, doi:10.3390/cancers13081924_

Round 1

Reviewer 1 Report

Kaufmann et al perform a statistical analysis to determine location of CD8+ cells with respect to hypoxic regions in squamous cell carcinoma of the head and neck. The work is rigorous, but some issues need to be addressed:

  • The quantification and analysis are done in specific ROIs within each sample. The criteria used to define these ROIs must be clearly explained. If it is not random, what features were used to ensure reproducibility?
  • Explain why CA9 was used as a measure of hypoxia instead of other markers such as HIF-1a. Explain advantages and disadvantages of each chosen marker (namely CA9 and CD8) and the criteria to choose them.
  • The pictures must be shown previous to the computational manipulation of the images to evaluate the staining and quality of the histopathological images of the sections.
  • The correlation or anti-correlation of T cells and hypoxia within solid tumors needs to be discussed in the Discussion section and/or explained in the Introduction

Reviewer 2 Report

     You have quantified and perhaps steamlined assessing that which is apparent visually. Is the association is important? You have no data to suppport or refute this. You demonstrated a better way to assess the realtionship.

Hence arguably this paper could be greatly shortened by concentrating only on methodology, without more than a very brief statement as to why you chose to look at this subject: clinically the topic is discussed in the literature  and you have a method to quantify the relationship better than visual judgment.
